# WEIGHT-BASED PERFORMANCE ESTIMATION FOR DIVERSE DOMAINS

## ABSTRACT

One of the limitations of applying machine learning methods in real-world scenarios is the existence of a domain shift between the source (i.e., training) and target (i.e., test) datasets, which typically entails a significant performance drop. This is further complicated by the lack of annotated data in the target domain, making it impossible to quantitatively assess the model performance. As such, there is a pressing need for methods able to estimate a model's performance on unlabeled target data. Most of the existing approaches addressing this train a linear performance predictor, taking as input either an activation-based or a performance-based metric. As we will show, however, the accuracy of such predictors strongly depends on the domain shift. By contrast, we propose to use a weight-based metric as input to the linear predictor. Specifically, we measure the difference between the model's weights before and after fine-tuning it on a self-supervised loss, which we take to be the entropy of the network's predictions. This builds on the intuition that target data close to the source domain will produce more confident predictions, thus leading to small weight changes during fine-tuning. Our extensive experiments on standard object recognition benchmarks, using diverse network architectures, demonstrate the benefits of our method, outperforming both activation-based and performance-based baselines by a large margin. Our code is available in an anonymous repository: `https://anonymous.4open.science/r/79E9/`

## 1 INTRODUCTION

Being able to estimate how well a trained deep network would generalize to new target, unlabeled datasets would be a key asset in many real-world scenarios, where acquiring labels is too expensive or unfeasible. When the training and target data follow the same distribution, this can easily be achieved by setting aside a validation set from the training data. However, such a performance estimator fails in the presence of a domain shift, i.e., when the target data differs significantly from the source one.

Recent studies (Deng & Zheng, 2021; Deng et al., 2021) address this by creating a meta-dataset incorporating multiple variations of the source data obtained by diverse augmentation techniques, such as background change, color variation, and geometric transformations, so as to mimic different domain shifts. Target datasets can then be sampled from this meta-dataset, and their ground-truth performance obtained by evaluating the source-trained network on them. In essence, this provides data to train a linear performance predictor, which in turn can be applied to the real target data.

The aforementioned studies differ in the quantities they use as input to this linear performance predictor. Specifically, Deng & Zheng (2021) rely on the Fréchet distance between the network activations obtained from the source samples and the target ones, whereas Deng et al. (2021) exploit the performance of the source network on the self-supervised task of rotation prediction. Unfortunately, while the resulting linear predictors perform well within the meta-dataset, their generalization to some real target datasets remains unsatisfactory, depending on the gap between the source and real target data. This is illustrated by the left plot of Fig.1, where the red point indicating the true performance on USPS lies far from the activation-based linear predictor shown as a black line.

In this paper, we therefore introduce the use of a completely different type of input to the linear predictor. Instead of using an activation-based or performance-based metric, we advocate the use of a weight-based one. This is motivated by recent studies showing that the network weights provide

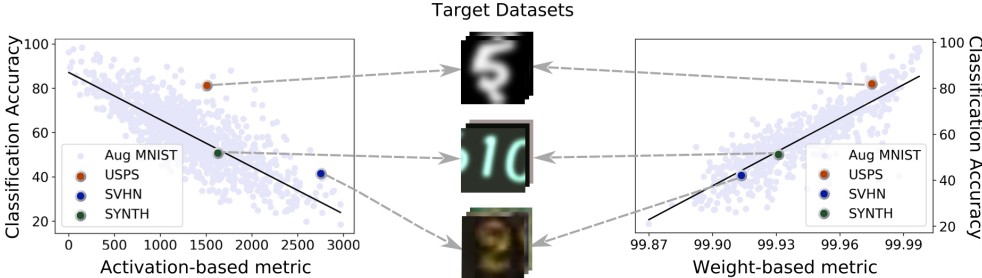

Figure 1: Correlation between classification accuracy and different metrics: Pearson correlation between network weights (right, our method) and Fréchet distance between network activations (left, (Deng & Zheng, 2021)). Note that our method yields a more reliable performance estimator, as evidenced by the points corresponding to the target datasets lying closer to the black line. The light-blue points correspond to sample sets from the meta-dataset.

valuable insights into model uncertainty (Lacombe et al., 2021), model complexity (Rieck et al., 2019), model compressibility (Barsbey et al., 2021), and in-domain generalization (Birdal et al., 2021; Franchi et al., 2022; Nagarajan & Kolter, 2019; Simsekli et al., 2020). Here, by contrast, we demonstrate the relationship between network weights and out-of-domain generalization. Specifically, we analyze how much the network weights change when fine-tuned on the target data with an unsupervised loss. This builds on the intuition that, the larger the domain gap between the source and the target datasets, the more the network will need to change to bridge this gap.

Computing our weight-based metric thus consists of two steps: Fine-tuning the last fully connected layers of the model with an unsupervised loss, and calculating the distance between the weights of the original model and those of the fine-tuned one. We use entropy minimization as an unsupervised loss, because of its convergence speed and of its independence from the model architecture; unlike other self-supervised losses, such as rotation prediction (Gidaris et al., 2018), the entropy is calculated directly on the model output, not requiring an additional network head.

In our experiments, we study two different weight-based distances: the Euclidean distance and the Pearson correlation. Our results evidence that both yield more reliable performance estimates than activation-based and performance-based ones. This is illustrated in the right plot of Fig. 1, where the points corresponding to the three real target datasets all lie close to the linear predictor. While alternative, more complex measures may also be viable, our work shows that even a basic norm-based approach surpasses other methods, which we evidence on several benchmark datasets and using different network architectures.

## 2 RELATED WORK

Existing methods can be categorized into activation-based and performance-based.

**Activation-based approaches** aim to find a criteria for performance estimation based on network activations. For example, Garg et al. (2022) propose Average Threshold Confidence (ATC) score based on the negative entropy of networks predictions. The authors acknowledge that ATC returns inconsistent estimates on certain types of distribution shifts. Another approach in this category (Schelter et al., 2020) explores various statistics derived from a prediction score. An alternative entropy-based method by Guillory et al. (2021) connects classification accuracy to the entropy difference in network activations between source and target data. However, its effectiveness relies on network calibration. Chen et al. (2021b) employ prior knowledge about the distribution shift to provide accurate performance estimates.

In contrast with the above-mentioned approaches that focus on the network output, Deng & Zheng (2021) analyze the feature representations. The authors propose to create a collection of augmented source datasets. They further learn a linear regression model to predict the accuracy on these sets based on the Fréchet distance between the source feature representations and the augmented feature representations. In our experiments, we observed that although there is a strong linear correlation between accuracy on the augmented source datasets and the Fréchet distance, real target datasets do not always follow the same pattern, thus leading to unsatisfactory accuracy estimates.

**Performance-based approaches** evaluate the classification accuracy of the network using its performance on self-supervised tasks. For instance, Deng et al. (2021) propose to learn a correlation between the rotation prediction accuracy and the classification accuracy. The works of Jiang et al. (2022); Chuang et al. (2020) show that the test error can be estimated by performing several trainings of the same network on the same source dataset, and measuring the disagreement rate between these networks on the target dataset. Building on this work, Chen et al. (2021a) learn an ensemble of models to identify misclassified points from the target dataset based on the disagreement between the models, and use self-training to improve this ensemble.

The aforementioned methods require access to the model during training. For example, in the work of Deng et al. (2021), the network architecture needs to be upgraded with the second head and trained on both tasks. The works of Jiang et al. (2022); Chuang et al. (2020); Chen et al. (2021a) require re-training of the source model to find the samples with disagreement. This might be undesirable for a large source dataset where training is time consuming. Note that our approach requires neither architecture alterations nor re-training on the source data.

In this work, we focus on analyzing the network weights, which was proven to be useful for various in-domain and out-of-domain tasks. For example Nagarajan & Kolter (2019) show that the distance of trained weights from random initialization is implicitly regularized by SGD and has a negative correlation with the proportion of noisy labels in the data. Hu et al. (2020) further use the distance of trained weights from random initialization as a regularization method for training with noisy labels. Yu et al. (2022) introduce a projection norm and show its correlation with out-of-distribution error.

By contrast, here, we study the relationship between a change in weights incurred from self-supervised fine-tuning and performance on the target data. Our approach compares favorably to the SOTA accuracy estimation methods from each of the above categories. We emphasize that our method requires neither prior knowledge of the nature of the distribution shift, nor target labels.

## 3    METHODOLOGY

Let us now introduce our approach to estimating how well a model trained on a source dataset would generalize to a target dataset from a different domain, in the absence of target supervision. Instead of predicting performance from the activation difference between the source and target samples or from the network performance on a different task, we propose to exploit the model's weights variations when fine-tuned with an unsupervised loss. Specifically, we consider the Euclidean distance and the Pearson correlation coefficient between the weights before and after fine-tuning, and empirically show that these metrics display a strong linear correlation with the model performance on the target task. We therefore learn this correlation with a linear regressor trained on augmented versions of the source data, which we use to predict the target data performance.

### 3.1    PROBLEM DEFINITION

Let $\mathcal{P}^S$ and $\mathcal{Q}^T$ be the probability distributions of the source and target domains, respectively, $\mathcal{D}_S : \{x_s, y_s\}^{n_s} \sim \mathcal{P}^S$ be a labeled source dataset with $n_s$ samples, and $\mathcal{D}_T : \{x_t\}^{n_t} \sim \mathcal{Q}^T$ be an unlabeled target dataset with $n_t$ samples. A model $f_\theta$ is trained on the source dataset $\mathcal{D}_S$ to predict a correct label: $f_\theta : x_i \to \hat{y}_i; x_i \sim \mathcal{D}^S$. Our goal then is to estimate the accuracy of the trained model $f_\theta$ on the unlabeled target dataset $\mathcal{D}_T$.

### 3.2    WEIGHT-BASED PERFORMANCE ESTIMATION

In this paper, we propose to predict model performance on target data based on its weight shift during unsupervised fine-tuning. This is motivated by the intuition that large domain gaps would lead to larger weight variations and also to lower accuracy than small domain gaps. Below, we first introduce our approach to measuring weight changes, and then present our accuracy predictor.

**Weight-based distance metrics.** Measuring a change in the model weights requires fine-tuning the model on the target data. In the absence of supervision, we propose to use the Shannon entropy. Given the model prediction $\hat{y} = f_\theta(x)$ encoded as a $C$-dimensional vector of class probabilities, the entropy can be written as $H(\hat{y}) = -\sum_{c=1}^{C} \hat{y}^c \log(\hat{y}^c)$ , where $\hat{y}^c$ is the probability for class $c$.

The entropy can be interpreted as an uncertainty of the model: Minimizing the entropy of the predictions encourages the network to produce confident predictions. Intuitively, if the target data is close to the source one, the model will produce confident predictions and will not significantly change during fine-tuning. Conversely, in the presence of a large domain shift, optimizing the entropy loss will result in a large network change.

To perform this comparison, we investigate the use of two distance measures. Specifically, given the weights $\theta$ before fine-tuning, and the weights $\hat{\theta}$ after fine-tuning, we compute the Euclidean distance and the Pearson correlation coefficient:

$$d_{eucl}(\theta, \hat{\theta}) = \sqrt{\sum_{i=0}^{n}(\theta_i - \hat{\theta}_i)^2} \ , \quad d_{prs}(\theta, \hat{\theta}) = \frac{\sum_{i=0}^{n}(\theta_i - \mu(\theta))(\hat{\theta}_i - \mu(\hat{\theta})))}{\sqrt{\sum_{i=0}^{n}(\theta_i - \mu(\theta))^2 \sum_{i=0}^{n}(\hat{\theta}_i - \mu(\hat{\theta}))^2}} \ ,$$

where $n$ is the number of weights considered, $\mu(\theta)$ is the mean of the weights.

**Fine-tuning the last layers for out-of-distribution adaptation.** Due to the high-dimensionality of the network weight space, comparing the network weights is non-trivial and may suffer from the curse of dimensionality. The impact of fine-tuning is not equally distributed across the network, with the last layers typically being affected more than the first ones (Kornblith et al., 2020). More importantly, fine-tuning the whole network distorts pretrained features when the distribution shift is large and therefore results in the model underperforming on target domains, as shown by Kumar et al. (2022). Therefore, by only updating the last layers, we expect a better direction for the performance improvements. As a result, we only fine-tune the classifier part of the network, consisting of all the fully connected layers at the end of the network, while freezing the feature extractor.

**Difference between the weights of fine-tuned and original networks, and its correlation to their performance gap.** Given $\theta^{(0)}$ - the weights of the network before fine-tuning; $\theta^{(k)}$ - the weights of the network at step $k$, S - the number of fine-tuning steps, $g_i^{(j)}$ - the gradient of the entropy loss at step $i$ w.r.t. $\theta^{(j)}$, and $\alpha$- the learning rate, our goal is to evaluate the quality and robustness of the extracted features w.r.t. the target dataset. Here, we explain how the two main aspects of the model updates, namely the magnitude and the consistency, are related to model generalizability.

- *Magnitude of the network updates:* The magnitude of the network modifications required to optimize an unsupervised loss function is encapsulated in its average gradient w.r.t. $\theta^{(0)}$, i.e., $\alpha \sum_{k \in 0}^{S} g_0^{(k)}$. The gradient's magnitude reflects the flatness of the loss function and can be regarded as an indicator of convergence (Zhang et al., 2023).

- *Consistency of the network updates.* The consistency of network updates across batches of the target dataset can be expressed through the coherence of gradients. We can quantify the consistency of the gradients between batch $i$ and $j$ by the derivative of their inner product, i.e., $\nabla_\theta(g_0^{(i)} \cdot g_0^{(j)}), i \neq j$. This inner product indicates the level of gradient alignment as well as the variance of the learned features for the target dataset (Guiroy et al., 2019). Specifically, if the inner product between the gradients is positive $(g_0^{(i)} \cdot g_0^{(j)}) \geq 0$, then updating the network weights along the direction of $g_0^{(i)}$ would enhance the performance on batch $j$, and vice versa. This means that strong gradient directions are indicative of training stability and, consequently, of the model's generalizability (Chatterjee, 2020).

Since computing $\nabla_\theta(g_0^{(i)} \cdot g_0^{(j)})$ requires calculating second-order derivatives, making it computationally prohibitive, we propose to approximate it in terms of the difference in weights before and after $k$ updates. As shown by Nichol et al. (2018) and Shi et al. (2022),

$$\mathbb{E}(\theta^{(0)} - \theta^{(k)}) = \alpha \sum_{k \in 1}^{S} g_0^{(k)} - \frac{\alpha^2}{s(s-1)} \sum_{i,j \in S}^{i \neq j} \nabla_\theta(g_0^{(i)} \cdot g_0^{(j)}) \ . \tag{1}$$

In the light of above discussion, we argue that the difference between the weights captures both the consistency and the magnitude of the network updates. Let us now further discuss the importance of consistent updates.

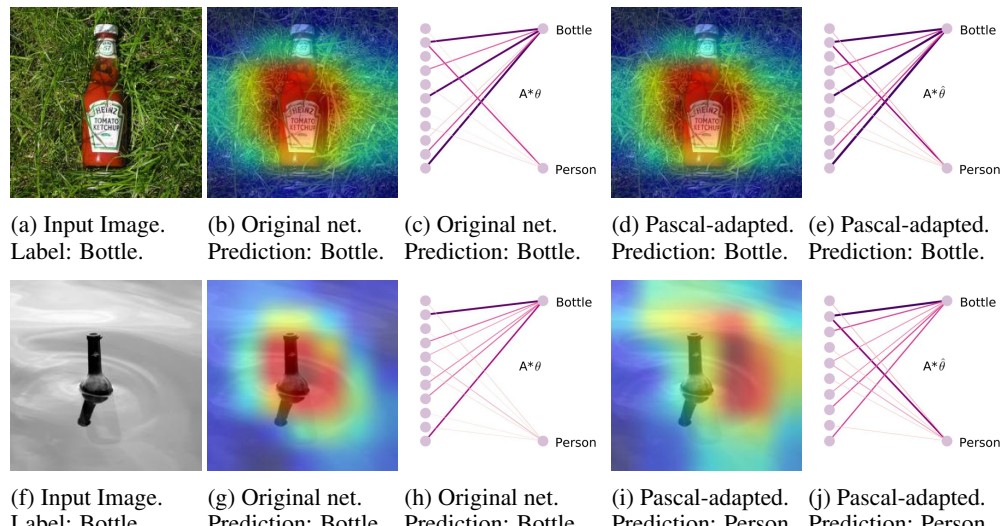

(a) Input Image. Label: Bottle.
(b) Original net. Prediction: Bottle.
(c) Original net. Prediction: Bottle.
(d) Pascal-adapted. Prediction: Bottle.
(e) Pascal-adapted. Prediction: Bottle.

(f) Input Image. Label: Bottle.
(g) Original net. Prediction: Bottle.
(h) Original net. Prediction: Bottle.
(i) Pascal-adapted. Prediction: Person.
(j) Pascal-adapted. Prediction: Person.

Figure 2: Class Activation Maps before and after fine-tuning. Top: Same predictions for the original and fine-tuned models. Bottom: The fine-tuned model's prediction differs from the original model.

Our underlying assumption is that the importance of the robust features contributing to each class should remain consistent across domains. We illustrate this by projecting the weights from the last layer onto the convolutional feature maps and generating the class activation maps (CAMs), as introduced by Zhou et al. (2015). CAMs highlight the discriminative regions of the image relevant to the prediction. For instance, for image $I_1$ in Fig.2a, the saliency map stays consistent after fine-tuning, with the network correctly identifying the class by focusing on the same region. However, for $I_2$ (Fig.2f), "Person" features dominate "Bottle" features, causing the network to shift attention to the background, as shown in Fig.2i.

The relationship between performance and weight changes becomes evident when examining the Hadamard product of activations and weights, shown in Figures 2c, 2h, 2e, and 2j. For images $I_1$ and $I_2$, the representations $A_1$ and $A_2$ are identical before and after fine-tuning due to the feature extractor being frozen. Thus, the robustness of the features w.r.t. the predicted classes is reflected in the change of $\hat{\theta}$ in relation to $\theta$.

**Accuracy predictor.** As illustrated by the right plot of Fig.1 and further evidenced by our experiments, there is a linear correlation between the network weight change after fine-tuning and the accuracy. In other words, the accuracy for a target dataset can be estimated using a linear regressor:

$$acc(f_\theta) = w_1 \cdot d(\theta, \hat{\theta}) + w_0 \ , \tag{2}$$

where $d$ is either the Euclidean distance $d_{eucl}$ or the Pearson correlation $d_{prs}$, and $w_0$ and $w_1$ are the trainable parameters of the linear regressor.

To train these parameters, we follow Deng & Zheng (2021) and create a meta-dataset consisting of a collection of datasets obtained by performing different augmentations of the source data. Specifically, a sample set $\hat{\mathcal{D}}_s^j$ in the meta-dataset is built as follows. First, a set of $m$ possible transformations $T = \{T_1, T_2, .., T_m\}$, corresponding to background change, geometric transformations, or color variations, is created. Then, $l$ images are randomly selected from the validation set $\{v_s\}$ of the source data, leading to a set $\{v_s^j\}^l \subset \{v_s\}$. A random selection of $t$ transformations $\tau = \{T_i\}_{i=1}^t$ is then applied to these images, resulting in the sample set $\hat{\mathcal{D}}_s^j = \tau[v_s^j]$. By repeating this process $k$ times, we create a collection of sample sets, which form the meta-dataset.

As each sample set originally comes from the source data, we can compute its true performance under model $f_\theta$. Similarly, we can fine-tune the model on each sample set using the entropy, and then compute the distance between the weights before and after fine-tuning. Altogether, this gives us supervised data, consisting of pairs of weight distance and true accuracy, from which we can learn the weights $w_0$ and $w_1$ of the linear regressor of Eq. 2.

### 3.3 Accuracy Prediction on Target Data

We can use the trained linear regressor to estimate the network performance on any unlabeled target dataset. Specifically, given a target dataset $\mathcal{D}_T : \{x_t\}^{n_t}$, we first split it into $k$ subsets of size $l$,

$$\mathcal{D}_t = \{\mathcal{D}_t^1, \mathcal{D}_t^2, ..., \mathcal{D}_t^k\}, \quad k = \left\lfloor \frac{n_t}{l} \right\rfloor,$$

so that the size of each subset matches the size of the validation sample sets.

Then, we fine-tune the network $f_\theta$ on $\mathcal{D}_t^j$, $\forall j \in [1, .., k]$ with our unsupervised entropy loss, and estimate the weight change using a distance measure. Given the obtained weight-based metric $d$, we use the trained linear regressor to predict the accuracy of $\mathcal{D}_t^j$ as $acc_j = w_1 \cdot d + w_0$. The final accuracy for the target dataset is calculated as the average accuracy of its subsets.

## 4 Experiments

We conduct extensive experiments on three benchmark datasets, Digits, COCO, and CIFAR10. For each dataset, we visualize the correlation between the accuracy and different metrics: The performance-based metric exploiting rotation prediction accuracy as in (Deng et al., 2021), the FID activation-based metric of Deng & Zheng (2021) that uses the Fréchet distance between the network activations, and our proposed weight-based metric. We further report the results of a linear regressor trained on either one of these metrics.

### 4.1 Datasets

**Digits** consists of a source domain, MNIST (LeCun et al., 2010), which contains 60K training and 10K test images, depicting grayscale handwritten digits distributed between 10 classes, and three target datasets: USPS (Denker et al., 1989), SVHN (Netzer et al., 2011), and SYNTH (Ganin & Lempitsky, 2015). The target datasets are also comprised of digit images of 10 classes, but with different colors, styles, and backgrounds. For this dataset, accuracy prediction is evaluated on two network architectures: LeNet (Lecun et al., 1998) and MiniVGG (aka VGG-7 (Simonyan & Zisserman, 2015)). Note that our results for the LeNet model differ from those reported in (Deng & Zheng, 2021), as we obtain significantly higher ground-truth accuracies with LeNet on all three target datasets.

**COCO**. Following Peng et al. (2018), we select a subset of the COCO dataset (Lin et al., 2014a) to build a source domain, with roughly 7K training and 6K validation samples, distributed in 12 categories. Our goal is to predict the accuracy of the models on 3 target datasets: Caltech (Griffin et al., 2007), Pascal (Everingham et al., 2010) and ImageNet (Deng et al., 2009), each sharing the same 12 classes with the source dataset. For this dataset, we use two network architectures: AlexNet (Krizhevsky et al., 2012) and ResNet50 (He et al., 2016). Given the COCO dataset's limited size, we employ pre-trained ImageNet weights and fine-tune the network on COCO.

**CIFAR10** contains one source domain, CIFAR10 (Krizhevsky & Hinton, 2009), with natural images from 10 classes, divided between 50K training samples and 10K test samples, and one target domain, CIFAR10.1 (Recht et al., 2018) with 2K test images. For this dataset, we employ a DenseNet(L=40, k=12) (Huang et al., 2017) architecture, where $L$ is the number of layers, and $k$ is the growth rate.

### 4.2 Baselines and Metrics

As mentioned before, we compare our approach to (Deng et al., 2021) and (Deng & Zheng, 2021). Additionally, we evaluate a baseline relying on the entropy score, which considers the prediction to be correct if its entropy is smaller than a certain threshold $\tau \in [0, 1]$. In other words, the prediction $\hat{y}$ is considered to be correct if $H(\hat{y}) \leq \tau * \log(C)$, where $C$ is the number of classes.

The last selected baselines are ATC (Garg et al., 2022) and COT (Lu et al., 2023). ATC improves the entropy score-based method by estimating the threshold from the validation set of the source data; COT employs the Earth Mover's Distance between labels from the source domain and predictions from the target domain.

Table 1: Results on Digits. MAE: Mean Absolute Error.

| | | LeNet | | | | MiniVGG | | | |
|---|---|---|---|---|---|---|---|---|---|
| | Metric Type | USPS | SVHN | SYNTH | MAE↓ | USPS | SVHN | SYNTH | MAE↓ |
| Ground Truth Accuracy | | 81.46 | 41.59 | 50.66 | | 84.55 | 38.16 | 48.38 | |
| Entropy Score $_{\tau=0.1}$ | A | 53.61 | 1.77 | 20.19 | 32.71 | 47.33 | 2.05 | 17.66 | 34.68 |
| Entropy Score $_{\tau=0.3}$ | A | 74.04 | 4.77 | 34.72 | 20.05 | 73.09 | 5.71 | 31.69 | 20.24 |
| ATC$_{val}$ | A | 69.11 | 3.89 | 30.8 | 23.30 | 72.70 | 5.54 | 31.06 | 20.60 |
| ATC$_{meta}$ | A | 99.25 | 30.87 | 86.1 | 21.34 | 93.82 | 18.04 | 61.46 | 14.16 |
| FID | A | 55.2 | 27.65 | 52.42 | 14.04 | 53.86 | 29.34 | 49.77 | 13.63 |
| COT | A | 74.14 | 26.38 | 54.39 | 8.76 | 73.91 | 31.09 | 54.54 | 7.96 |
| Rotation accuracy | P | 53.01 | 29.31 | 31.46 | 20.09 | 60.45 | 29.48 | 31.43 | 16.58 |
| Euclidean Distance | W | 73.44 | 44.64 | 51.75 | 4.06 | 76.23 | 37.13 | 52.83 | **4.60** |
| Pearson Correlation | W | 73.65 | 43.48 | 52.58 | **3.87** | 76.24 | 31.36 | 53.12 | 6.62 |

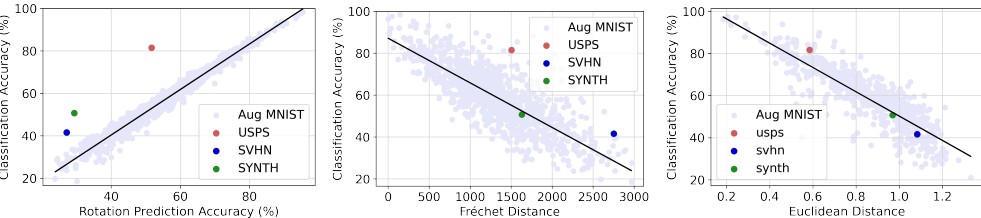

Figure 3: Correlation between the classification accuracy of LeNet and various metrics: P: Performance-based (left), A: Activation-based (middle), W: Weight-based (right, our method).

For a fair comparison, we estimate the ATC threshold from either the validation data (ATC$_{val}$ (Garg et al., 2022)) or the meta-dataset (ATC$_{meta}$). Our experiments show that COT$_{val}$ (Lu et al., 2023) outperforms COT$_{meta}$ across all datasets; hence, we omit the latter in our analysis.

For each dataset, we report the true accuracy obtained on the target data, the accuracy predicted by a performance prediction method, and the mean absolute error (MAE) between these two values, averaged over the different target sets in each dataset.

## 4.3 EXPERIMENTAL RESULTS

**Results on Digits.** Let us start with the discussion of the criteria for assessing the effectiveness of accuracy prediction. In particular, we discovered that it is not sufficient to evaluate accuracy prediction based on the correlation between the input metric (performance-based, activation-based, or weight-based) and the accuracy only within the meta-dataset, as it does not necessarily reflect the correlation with the target datasets. We illustrate this with the plots in Figure 3.

Specifically, the plots reveal that, within the meta-dataset, the performance-based metric exhibits the largest correlation with classification accuracy. However, all three target datasets are located far from the main trend of the meta-dataset. Consequently, the performance of the linear regression model trained with rotation prediction accuracy as input is unsatisfactory for this setup, with more than 28% gap between the ground-truth accuracy and the predicted accuracy for the USPS target dataset, as shown in Table 1.

While the Fréchet distance between activations has a smaller correlation with the classification accuracy within the meta-dataset, it yields more accurate predictions on the target datasets. It is especially evident when predicting the performance of both LeNet and MiniVGG for SYNTH dataset (see FID in Table 1). This evidences that there is a trade-off between a high correlation withing the meta-dataset and generalizing to diverse target domains.

Our proposed weight-based approach satisfies this requirement. As shown in Figure 3, the Euclidean distance between the weights of pre-trained and fine-tuned models is correlated with the classification accuracy not only within the meta-dataset, but also for the target ones. This is also evidenced by the numbers in Table 1, showing that the predictions produced by our linear regressor for the target datasets are more precise than those of the other approaches, with only $4.06\%$ average absolute error for LeNet, and $4.6\%$ for MiniVGG. A similar trend can be observed when using the Pearson correlation to estimate the difference between fine-tuned and pre-trained models; this metric can therefore be used as an alternative to the Euclidean distance.

Table 2: Results on COCO. MAE: Mean Absolute Error.

| | | AlexNet | | | | ResNet50 | | | |
|---|---|---|---|---|---|---|---|---|---|
| | Metric Type | Caltech | Pascal | ImageNet | MAE ↓ | Caltech | Pascal | ImageNet | MAE ↓ |
| Ground Truth Accuracy | | 86.89 | 71.87 | 78.00 | | 94.10 | 85.92 | 92.77 | |
| Entropy Score ($\tau = 0.1$) | A | 60.23 | 50.76 | 53.0 | 24.26 | 60.49 | 47.33 | 56.17 | 36.27 |
| Entropy Score ($\tau = 0.3$) | A | 81.65 | 71.55 | 73.33 | 3.41 | 86.75 | 73.03 | 80.83 | 10.73 |
| $ATC_{val}$ | A | 88.86 | 79.82 | 80.67 | 3.66 | 90.74 | 80.23 | 82.33 | 6.50 |
| $ATC_{meta}$ | A | 89.57 | 81.18 | 81.9 | 4.20 | 92.25 | 82.00 | 84.20 | 4.77 |
| FID | A | 57.63 | 63.97 | 67.85 | 15.77 | 78.12 | 82.45 | 85.88 | 8.77 |
| COT | A | 58.98 | 68.30 | 77.38 | 10.70 | 62.60 | 69.61 | 85.70 | 18.11 |
| Rotation accuracy | P | 100.00 | 83.48 | 90.55 | 12.42 | 92.40 | 90.20 | 90.48 | 3.16 |
| Euclidean Distance | W | 87.11 | 79.0 | 79.65 | **3.00** | 91.55 | 85.81 | 93.68 | **1.19** |
| Pearson Correlation | W | 84.25 | 78.31 | 78.87 | 3.32 | 88.70 | 85.78 | 89.92 | 2.79 |

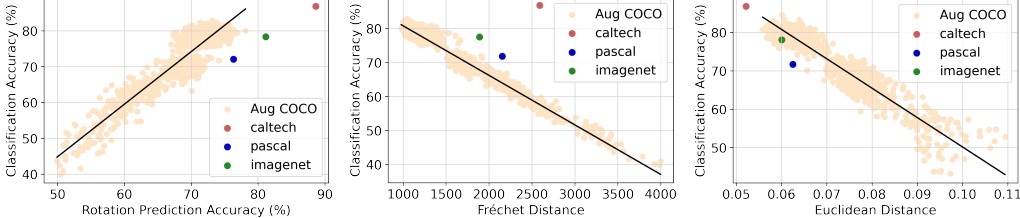

Figure 4: Correlation between the classification accuracy of AlexNet and various metrics:
P: Performance-based (left), A: Activation-based (middle), W: Weight-based (right, our method).

**Results on COCO.** Here, we focus on predicting the performance of deeper networks, e.g., AlexNet and ResNet50, on 3 target datasets: Caltech, Pascal, and ImageNet.

The plots from Figure 4 reveal a similar trend to the Digits dataset for the linear correlation of the activation- and performance-based metrics within the meta-dataset, which does not necessarily persist for the target domains. Caltech is the most segregated domain, with both baselines predicting accuracy with more than a 10% absolute error. Interestingly, the Entropy-Score based method with $\tau = 0.3$ outperforms the other baselines for AlexNet, yet selecting the right $\tau$ is not a trivial task. For example, the same criterion on ResNet50 produces unsatisfactory results on all target datasets.

Differently from the Digits setup, the ATC-based approach for the COCO setup yields a substantial improvement over the entropy score-based one and even outperforms FID. Note that both networks generalize well on the target domains (e.g., the lowest ground-truth accuracy is 71.9% for Digits vs 38.16% for COCO), which results in more confident predictions, and therefore more precise entropy-based estimation. However, the ATC performance remains inferior to ours. Finally, we discover that COT yields poor predictions, with the mean error reaching 18% for ResNet.

Unlike other metrics, our approach generalizes well across all target datasets. Our linear regressor outperforms the other approaches by a large margin, with an MAE of $3\%$ for AlexNet and $2,79\%$ for ResNet. Additionally, our method can successfully predict performance for the Caltech dataset, where the baseline metrics fail, with an MAE of just $0.21\%$ for AlexNet, and $2.54\%$ for ResNet50.

**Results on CIFAR10.** We conclude our analysis with experiments on the CIFAR10 dataset. The resulting prediction accuracies are shown in Table 3. We first note that, for the CIFAR10 setup, the best fixed entropy score is defined by $\tau = 0.1$, while for the COCO setup, the best prediction was achieved with $\tau = 0.3$. This observation confirms that the optimal entropy threshold varies depending on the domain. In comparison, the ATC-based method provides more accurate predictions on both setups, with just $1.5\%$ MAE for CIFAR10.1.

The other activation-based method FID gives worse accuracy prediction than the entropy-based methods. Similarly to the previous setups, we observe the issue of the FID for the target dataset lying away from the main trend of the augmented source datasets. Notably, the last activation-based method COT provides almost exact performance prediction for the CIFAR10.1 dataset. This accuracy is attributed to the calibration of the model, facilitated by the resemblance between the validation set and the target set. However, as was stated by the authors (Lu et al., 2023) and confirmed by our experiments in the previous sections, for more complex natural distribution shifts COT overestimates the error due to the model providing less confident predictions on OOD samples.

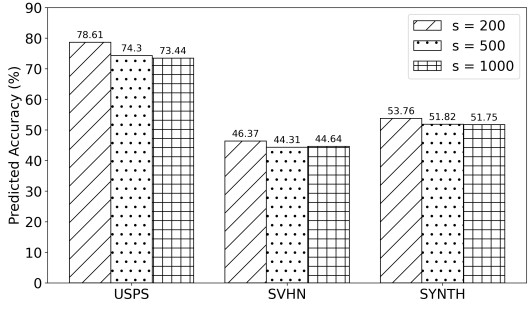

Figure 5: Correlation between the classification accuracy of DenseNet and various metrics: Performance-based (left), Activation-based (middle), Weight-based (right, our method).

|  | CIFAR10, DenseNet | |
|---|---|---|
|  | CIFAR10.1 | MAE |
| Ground Truth Acc | 88.65 | |
| Entropy ($\tau = 0.1$) | 85.50 | 3.25 |
| Entropy ($\tau = 0.3$) | 99.55 | 10.90 |
| $ATC_{val}$ | 87.15 | 1.50 |
| $ATC_{meta}$ | 90.80 | 2.15 |
| FID | 97.55 | 8.90 |
| COT | 88.75 | **0.10** |
| Rotation | 91.43 | 2.78 |
| Euclidean Distance | 89.63 | 0.98 |
| Pearson Correlation | 85.50 | 3.15 |

Table 3: Results on CIFAR10.

Figure 6: Sensitivity of our predictor to the target set size for Digits setup, evaluated on LeNet.

The performance-based approach demonstrates that the linear correlation between the rotation prediction accuracy and the classification accuracy persists for the target dataset, with a resulting accuracy estimate achieving 2.78% MAE.

Finally, we show that for the CIFAR10 setup, our weight-based method using the Euclidean distance outperforms most of the baselines, with less than 1% MAE from the ground-truth. The Pearson correlation metric, however, does not perform as well to predict the accuracy of CIFAR10.1, due to its non-linear distribution with respect to the classification accuracy. Across all experiments, we notice that the Euclidean distance is generally more stable than the Pearson correlation.

**Sensitivity to the Target Set Size.** Finally, we show that our method scales to scenarios with limited access to the test data, where only a small number of unlabeled test samples is available for evaluation. To confirm this, we use the Digits setting with the LeNet backbone. We split the target datasets of into chunks of size $k$, with $k \in [200, 500, 1000]$, and use our approach to predict the accuracy for each split.

The barplot in Figure 6 shows that the predicted accuracy for all the target datasets does not significantly change when using only 500 samples, with the average MAE over the target datasets marginally increasing from 4.06% to 4.32%. However, further reducing the dataset size negatively affects the performance of our method and further increases the MAE to 5.3%. Nevertheless, even with the smallest sample size of 200, our accuracy predictor outperforms the baselines that use the complete target datasets.

For a comprehensive ablation analysis on every stage of our pipeline, encompassing the impact of the unsupervised task, the metric, and the representative layer, we direct the readers to the Appendix.

# 5 CONCLUSION

In this work, we have tackled the problem of predicting the performance of a network on unlabeled target data whose distribution differs from that of the source training data. To this end, we have proposed a new weight-based approach that estimates the performance of the network from the degree of weight changes incurred by fine-tuning the network on the target dataset with an unsupervised loss. Our extensive experiments have shown that our approach effectively predicts the accuracy across a variety of domain shifts and network architectures. Note that, as performance-based predictors, our approach requires fine-tuning on the target data. In the future, we will investigate if this process can be sped up by restricting the number of iterations or of fine-tuned weights.

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

APPENDIX

## A   UNDERSTANDING WEIGHT DIFFERENCES THROUGH GRADIENTS

In this section, we provide a detailed analysis of weight differences from the perspective of gradients.

We assume the following notations: $\theta^{(0)}$ - the weights of the network before fine-tuning, $\theta^{(k)}$ - the weights of the network at step $k$, S - the number of fine-tuning steps, $g_i^{(j)} = \nabla L^{(i)}(\theta^{(j)})$ - the gradient of the unsupervised entropy loss at step $i$ w.r.t. $\theta^{(j)}$, $H_j^{(i)} = \nabla_\theta^2 L^{(i)}(\theta^{(j)})$ - the Hessian at step $i$ w.r.t. $\theta^{(j)}$, and $\alpha$- the learning rate.

After one step of gradient descent, the difference between the weights is a gradient at step 0:

$$\theta^{(0)} - \theta^{(1)} = \alpha \nabla_\theta L^{(0)}(\theta^{(0)}) = \alpha g_0^{(0)} \tag{3}$$

After the second gradient descent step, the distance between the weights has an additional gradient $g_1^{(1)}$, calculated at step 1 w.r.t. the updated weights $\theta^{(1)}$ :

$$\theta^{(0)} - \theta^{(2)} = \alpha g_0^{(0)} + \alpha g_1^{(1)} \tag{4}$$

Following the works of Nichol et al. (2018) and Shi et al. (2022), we can approximate $g_1^{(1)}$ using First-order Taylor series as follows : $g_1^{(1)} = g_0^{(1)} - \alpha H_0^{(1)} g_0^{(0)} + \mathcal{O}(\alpha^2)$. Plugging it back into 4:

$$\theta^{(0)} - \theta^{(2)} = \alpha(g_0^{(0)} + g_0^{(1)}) - \alpha^2 H_0^{(1)} g_0^{(0)} + \mathcal{O}(\alpha^3) \tag{5}$$

Since the batches are drawn randomly,

$$\mathbb{E}_{0,1}(H_0^{(1)} g_0^{(0)}) = \frac{1}{2}\mathbb{E}_{0,1}(H_0^{(1)} g_0^{(0)} + H_0^{(0)} g_0^{(1)}) = \frac{1}{2}\mathbb{E}_{0,1}(\nabla_\theta(g_0^{(0)} \cdot g_0^{(1)}))$$

Plugging it back into 5:

$$\theta^{(0)} - \theta^{(2)} = \alpha(g_0^{(0)} + g_0^{(1)}) - \frac{\alpha^2}{2}\nabla_\theta(g_0^{(0)} \cdot g_0^{(1)}) + \mathcal{O}(\alpha^3) \tag{6}$$

Here the first term represents the average gradient of the unsupervised entropy loss $L$. The second term contains the dot product between gradients of two batches, which summarizes gradient alignment and shows the invariance of the learned features for the target dataset.

After $k$ gradient descent steps:

$$\mathbb{E}(\theta^{(0)} - \theta^{(k)}) = \alpha \sum_{k \in 1}^{S} g_0^{(k)} - \frac{\alpha^2}{S(S-1)} \sum_{i,j \in S}^{i \neq j} \nabla_\theta(g_0^{(i)} \cdot g_0^{(j)}) \tag{7}$$

Therefore, the norm of the difference between original and fine-tuned weights encapsulates both the extent of the network change (average gradient) and the consistency of this change within the target dataset (the degree of gradient alignment between batches).

## B   ABLATION STUDY

In this paper, we propose an approach for estimating the performance of a model on unlabeled target domains in the presence of domain shift. The main steps of the proposed pipeline are summarized in Figure 7. We first study the robustness of our method to the size of the target dataset. Next, we closely examine each step of the proposed pipeline, and explain the choice of the selected methods and metrics.

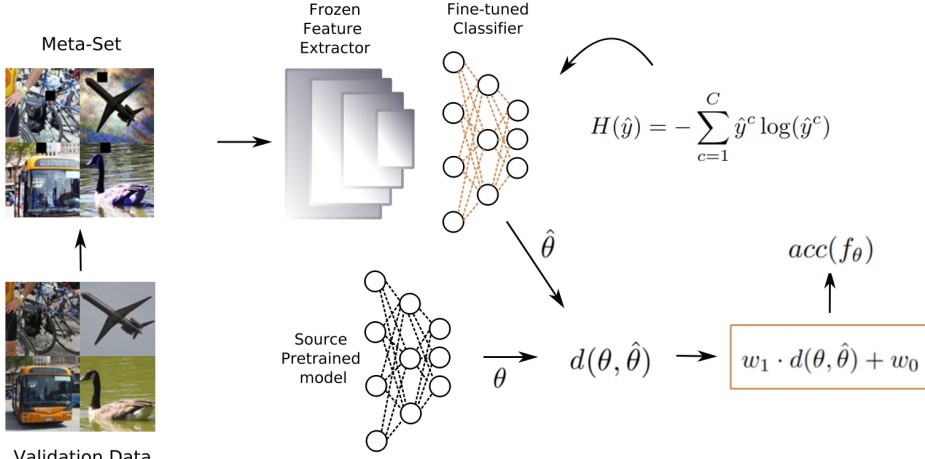

Figure 7: Weight-based performance estimation. We first construct a meta-dataset by augmenting the validation subset of the source data. The model is then fine-tuned on the meta-dataset with an unsupervised entropy loss. We compute the difference in weights between the fine-tuned model and the source pre-trained one. Finally, we train a linear regression model to predict the accuracy of the meta-dataset from the weight-based distance.

Table 4: Results on Wilds Benchmark. MAE: Mean Absolute Error

|  | Camelyon | IWildCam | FMoW | MAE |
|---|---|---|---|---|
| Ground Truth Accuracy | 72.91 | 67.69 | 52.90 | |
| Entropy Score ($\tau = 0.1$) | 18.39 | 66.75 | 27.56 | 26.94 |
| Entropy Score ($\tau = 0.3$) | 53.6 | 89.1 | 52.46 | 13.72 |
| $ATC_{val}$ | 84.22 | 66.03 | 55.27 | 5.11 |
| FID | 71.22 | 76.08 | 57.79 | 4.92 |
| COT | 75.86 | 56.01 | 52.59 | 4.98 |
| Euclidean Distance | 74.35 | 71.57 | 52.3 | 1.97 |

## B.1 ADDITIONAL EXPERIMENTS: WILDS BENCHMARK

To address a broaderspectrum of distribution shifts, we incorporate three additional datasets from the Wilds benchmark (Koh et al., 2021) into our experimental setup, namely Camelyon17 (Bándi et al., 2019), iWildCam (Beery et al., 2020) and fMoW (Christie et al., 2018).

• The Camelyon17 dataset contains patches from Whole-Slide images (WSI) potentially indicating metastatic breast cancer. The training set comprises 30 WSIs from three hospitals, while the test split consists of 10 WSIs from another hospital. The shift in this dataset reflects variations in data collection and processing between hospitals.

• The iWildCam dataset focuses on domain generalization in wildlife monitoring, utilizing camera traps to classify 182 animal species in photos from various traps. The shift in this dataset encompasses variations in illumination, angle, background, vegetation, color, and animal frequencies across traps. The division into train and test splits is determined based on the trap locations.

• The fMoW dataset includes satellite images categorized into 62 building or land classes. The separation into train and test sets is based on the years the images were taken, with training comprising images taken before 2012 and testing including images captured after 2016.

The results are presented in Table 4. Note that we omit the results for Rotation Prediction, as it proves ineffective in realistic shifts. For instance, predicting rotation in WSIs within the Camelyon dataset is unfeasible. The results show the superiority of our approach over the other baselines and prove its viability for a diverse range of natural distribution types.

## B.2 SENSITIVITY TO THE TARGET SET SIZE

In this section, we examine the sensitivity of our accuracy predictor to the target size for the COCO (Lin et al., 2014b) and CIFAR10 (Krizhevsky & Hinton, 2009) target datasets. In comparison with the Digits dataset (LeCun et al., 2010), presented in Section 4.7 of the main paper, the source classifiers of COCO and CIFAR10 generalize better on the target domains, with accuracies for all the target datasets exceeding 75%.

To create scenarios where only a limited number of target samples is available, we split the target datasets into batches of size $s$, and predict the accuracy for each batch. The accuracy prediction for each target dataset, averaged across all the batches, is displayed in Figure 8. According to the results, decreasing the size of the target set to 500 samples does not significantly change the accuracy prediction: for all the target datasets in both setups, the predictions for $s \geq 500$ do not vary by more than 2%. This means that our method is able to accurately estimate the classification performance for a domain with only 500 samples.

However, further decreasing the size of the target dataset leads to a deterioration of the prediction accuracy, with an MAE of 5.15% for the COCO target datasets and of 3.63% for CIFAR10.1. We therefore suggest to use our approach for datasets with at least 500 samples.

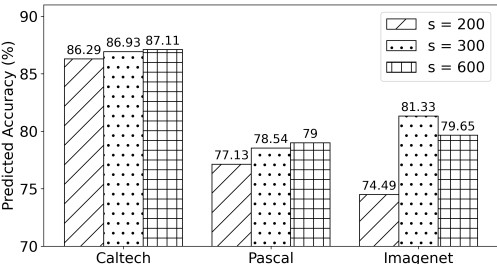 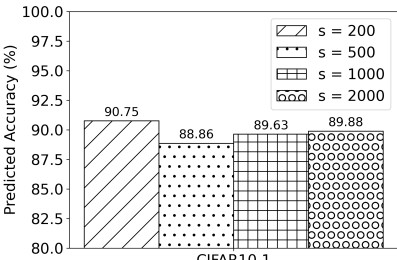

Figure 8: Predicted accuracy for target datasets from AlexNet trained on COCO (left) and DenseNet trained on CIFAR10 (right)

## B.3 CHOICE OF THE UNSUPERVISED TASK

In the proposed approach, the unsupervised loss was chosen to be entropy minimization. In this section, we discuss its advantages over other unsupervised tasks. We first show that the entropy value is a fair estimator of the model's accuracy, and without fine-tuning, the entropy value exhibits a correlation with the performance of the model. Intuitively, the model that is trained on the source data should be more uncertain on the target data when the domain shift is large, which corresponds to a large entropy value.

To validate this hypothesis, we perform the following experiment: We first create a meta-dataset by augmenting the source dataset, similarly to the first step of our proposed approach. Then, we plot the value of the entropy against the classification accuracy for the meta-dataset and for each target dataset. The resulting plots for each experimental setup (Digits, COCO and CIFAR10) are summarized in Figure 9 (top). The plots reveal a strong linear correlation between the entropy value and the classification accuracy within the meta-dataset. Therefore, similarly to the main experiments, a linear regressor can be trained to predict the performance on the target domains.

The results in Table 5 show that for most target domains, the entropy value is more representative of the classification performance than the rotation prediction accuracy. Additionally, choosing entropy minimization exempts us from modifying the training process of the model on the source data. These results confirm our choice of entropy minimization over rotation prediction.

However, using only the entropy as a base for performance prediction on diverse domains does not provide satisfactory results, with only 7.64 % MAE for Digits with MiniVGG, and 8.8 % MAE for COCO with AlexNet. Our weight-based metric yields significantly better results.

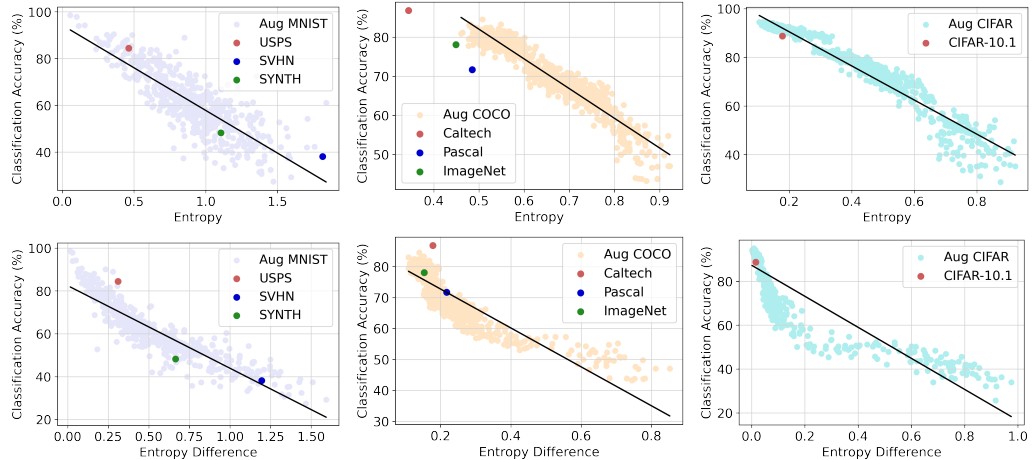

Figure 9: Correlation between the classification accuracy with the entropy (top) and the entropy difference before and after fine-tuning (bottom) for Digits with MiniVGG (left), COCO with AlexNet (middle), and CIFAR10 with DenseNet (right).

Table 5: Ablation Study: Metric choice. We report the Absolute Error between the predicted accuracy and the ground-truth accuracy. MAE-Mean Absolute Error.

|  | Digits, MiniVGG | | | | COCO, AlexNet | | | | CIFAR10 |
|  | USPS | SVHN | SYNTH | MAE↓ | Caltech | Pascal | ImageNet | MAE↓ | CIFAR10.1 |
|---|---|---|---|---|---|---|---|---|---|
| Entropy Value | 7.26 | 10.02 | 5.63 | 7.64 | 7.09 | 11.47 | 7.83 | 8.8 | 3.41 |
| Rotation Prediction | 24.24 | 8.68 | 16.92 | 16.61 | 13.09 | 11.61 | 10.97 | 11.89 | 2.75 |
| Entropy Difference | 13.92 | 1.77 | 8.59 | 8.09 | 12.68 | 0.04 | 2.39 | 5.04 | 2.47 |
| Euclidean Distance | 8.32 | 1.03 | 4.45 | 4.60 | 0.22 | 7.13 | 1.65 | 3.00 | 0.98 |

## B.4 CHOICE OF THE METRIC

To improve the performance of the accuracy predictor, we propose to fine-tune the network with the selected unsupervised loss, and estimate the domain gap by analyzing the degree of model change. In our approach, we focus on the difference between the weights of the source model and those of the fine-tuned model.

An alternative metric to the weight difference might be the entropy difference: seeing that the entropy value before fine-tuning is correlated with the accuracy (as shown in the previous section), it is natural to assume that after fine-tuning on the target dataset, the degree of the entropy change might be correlated with the classification performance. In other words, if the entropy difference is small, it implies that the fine-tuning did not cause significant change in the model predictions.

To validate this, we fine-tune the network for 2 epochs and study the degree of the entropy change before and after fine-tuning. The experimental results in Figure 9 (bottom) confirm the existence of a correlation between the entropy difference and the classification accuracy; however, it does not have the same linear trend as the entropy value. As a result, the linear regressor trained on the entropy difference, does not provide accurate performance estimates on all target datasets, as shown in Table 5.

Unlike entropy difference, the weight difference satisfies both desired criteria: it has a linear correlation with the classification accuracy within the meta-dataset; and it exhibits a similar behavior on the target datasets.

Table 6: Ablation Study: Layer choice. We report the Absolute Error between the predicted accuracy and the ground-truth accuracy. MAE-Mean Absolute Error.

| | Digits, MiniVGG | | | | COCO, AlexNet | | | |
|---|---|---|---|---|---|---|---|---|
| | USPS | SVHN | SYNTH | MAE ↓ | Caltech | Pascal | ImageNet | MAE ↓ |
| Euclidean Distance, 1-st layer | 8.32 | 1.03 | 4.45 | 4.38 | 0.22 | 7.13 | 1.65 | 3.00 |
| Euclidean Distance, 2-d layer | 8.68 | 1.69 | 2.84 | 4.40 | 1.93 | 6.14 | 1.72 | 3.26 |
| Euclidean Distance, 3-d layer | 10.70 | 7.54 | 4.10 | 7.45 | 3.16 | 4.68 | 15.19 | 7.68 |

## B.5 Choice of the Layer

To conclude our ablation study, we examine the choice of a representative layer, which would best reflect how the model changes from fine-tuning on the target dataset. Based on our proposed approach, all the layers of the classification head should be updated during fine-tuning. Then, if the classification head consists of several layers, one layer is selected for comparison; the final accuracy is estimated based on the weight change of this representative layer. In this section, we show that the first few layers of the classification head can be used for representing the model evolution during fine-tuning.

We select two models from our experimental setup, having more that one layer in their classification heads: MiniVGG for the Digits dataset and AlexNet for the COCO dataset. The classification head is first fine-tuned on the target datasets. Then, we separately select each layer as a representative layer to predict the performance of the source classifier. The results, summarized in Table 6, show that choosing the first fully-connected layer of the classifier as a representative layer results in better accuracy prediction for some datasets, e.g., USPS in Digits, and Caltech in COCO, whereas choosing the second fully-connected layer leads to more accurate predictions on other datasets, e.g., SYNTH in Digits and Pascal in COCO. However, overall, the Mean Absolute Error across the target datasets does not significantly vary when choosing the first or the second layer, with 0.02% MAE difference for Digits and 0.26% MAE difference for COCO. Choosing the last layer, however, results in the worst performance for both setups, with 7.45 % MAE for Digits and 7.68% for COCO.

Table 7: Ablation Study: Projection Norm vs Entropy Minimization. We report the Absolute Error between the predicted accuracy and the ground-truth accuracy. MAE-Mean Absolute Error.

| | Digits, LeNet | | | | COCO, AlexNet | | | |
|---|---|---|---|---|---|---|---|---|
| | USPS | SVHN | SYNTH | MAE ↓ | Caltech | Pascal | ImageNet | MAE ↓ |
| Our approach | 8.03 | 3.06 | 1.09 | 4.06 | 0.22 | 7.13 | 1.65 | 3.00 |
| Projection Norm (Yu et al., 2022) | 10.35 | 8.31 | 6.31 | 8.32 | 13.30 | 15.53 | 23.90 | 17.57 |

Finally, we compare our method to a weight-based approach of Projection Norm (Yu et al., 2022), applied on analyzing out-of-distribution (OOD) error. Our method deviates from (Yu et al., 2022) in a sense that our method tackles a more complex task of performance prediction, while the work of Yu et al. (2022) studies the correlation of their measure with the OOD test error. However, we can adapt Projection Norm to our problem by adding a meta-dataset creation and training a linear regression steps. The results, outlined in Table 7, show that Projection norm is inefficient in the presence of a large domain shifts due to the inaccuracy of generated pseudo-labels and a slow speed of convergence.

## C  Training Hyper-parameters

In this section, we summarize the hyperparameters used in our experiments.

**Source training hyper-parameters:** LeNet and MiniVGG are trained from scratch on MNIST for 30 epochs, using the Adam optimizer with a learning rate of 0.001.

AlexNet and ResNet50 are pre-trained on ImageNet and fine-tuned on the COCO dataset. AlexNet is fine-tuned for 30 epochs with the Adam optimizer and a learning rate of 0.001; ResNet50 is fine-tuned for 50 epochs with SGD with a Nesterov momentum of 0.9 and a learning rate of 0.001.

DenseNet is trained on CIFAR10 with SGD with a Nesterov momentum of 0.9 and a learning rate of 0.1. The network is trained from scratch for 200 epochs.

We use Densenet-121 (Huang et al., 2017) for the fMoW and Camelyon17 datasets, and ResNet50 (He et al., 2016) for the iWildCam dataset, and follow the ERM training procedure from the WILDs benchmark (Koh et al., 2021) for the main training. We use in-distribution validation splits as the validation data for each experiment, and out-of-distribution test split for performance prediction.

**Fine-Tuning hyperparameters:** During fine-tuning, for all the networks, we use SGD with a Nesterov momentum of 0.9, a weight decay of 0.0001 and a batch size of 64. Additionally, LeNet, MiniVGG and DenseNet are fine-tuned for 2 epochs with a learning rate of 0.01; AlexNet is fine-tuned with a learning rate of 0.001. For Wilds benchmark, the classifier part of each network is fine-tuned for 2 epochs, using SGD with lr= 0.0001 for fMoW and iWildCam, and lr= 0.01 for Camelyon.

Based on the size of target data available, we adapt the subset size $l$, described in Section 3.3, as follows: $l = 1000$ for the Digits dataset, $l = 600$ for the COCO dataset, and $l = 2000$ for CIFAR10. Note that we also provide an ablation study for the sensitivity of our method to the size $l$ in Section 4.7 of the main paper and in Section 1.1 of the Appendix.

**Meta-Dataset Augmentations:** As MNIST contains grayscale images, we perform diverse background changes to generate a wide range of augmentations. Specifically, we create binary masks from the MNIST samples. We then select a test sample from the COCO dataset, and mine patches to match the size of the binary masks. Finally, we invert the values of the patches in the location of the MNIST binary masks. A sample of MNIST augmentations can be found in Figure 10.

For the COCO and CIFAR10 datasets, we use the RandAugment (Cubuk et al., 2020) automated augmentation strategy. For each sample set, we randomly select an augmentation magnitude and three transformations from the following pool of transformation types: `cutout`, `auto_contrast`, `contrast`, `brightness`, `equalize`, `sharpness`, `solarize`, `color`, `posterize`, `translate_x`, `translate_y`. A sample of the described augmentations can be found in Figure 11. Note that RandAugment includes both geometric and color transformations, and is fully automated. Differently from (Deng & Zheng, 2021), we do not apply a computationally expensive background replacement on the COCO dataset. In fact, we show that even with these simple transformations, our approach is able to capture a variety of domain shifts.

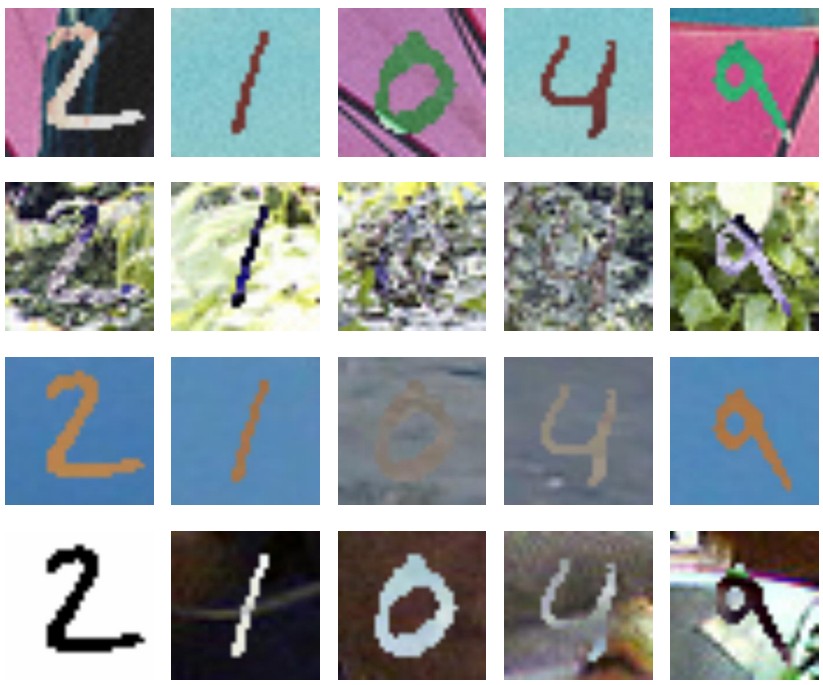

Figure 10: Meta-Dataset Digits.
Each row represent images from the same Sample Set.

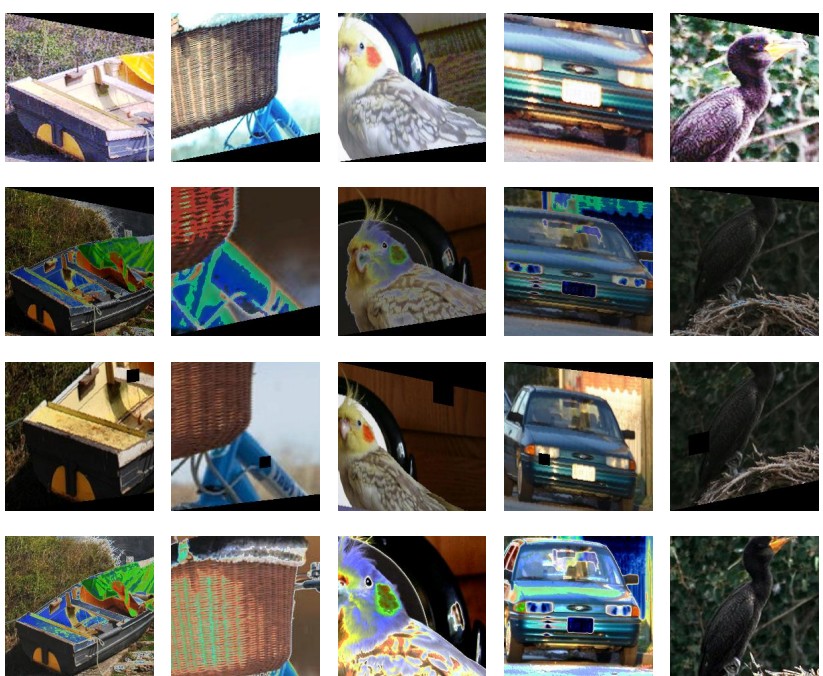

Figure 11: Meta-Dataset COCO.
Each row represent images from the same Sample Set.

