# OpenReview forum: "Weight-Based Performance Estimation for Diverse Domains"
_ICLR.cc/2024/Conference — Submitted to ICLR 2024_

### Official Review · Reviewer_PWrH · 2023-10-28

**Soundness:** 2 fair
**Presentation:** 3 good
**Contribution:** 2 fair
**Rating:** 6
**Confidence:** 3

**Summary:**

The authors tackle the practical and challenging problem of estimating performance of a given trained model on new data that is unlabeled and could be out of distribution from the model’s original training and test data. Making progress on this problem has the potential to save downstream users of machine learning models significant time and effort implementing an ML-appropriate data labeling process to evaluate performance of a model, only to determine that existing models do not perform well enough to be useful for their task of interest. This has the downside of making that labeling process a significant investment of resources without clear benefit, which is increasingly harmful if the users have limited resources, for example nonprofits or historically under-resourced communities. The authors propose an alternate approach for unlabeled performance estimation that is designed to be more predictive when domain shifts occur between the training dataset and the unlabeled test data. This method utilizes a self-supervised finetuning step on the unlabeled dataset and looks at the amount of weight change in the network activations pre and post finetuning as the input to a linear predictor of performance, based on the idea that a smaller change in model weights would correspond to more similarity between train and the unlabeled dataset and thus higher performance. They show that this method outperforms previous performance estimation methods focused on linear prediction from activations or predicted scores on some types of domain shift.

**Strengths:**

The paper is clearly written, the related work section is clear and their contributions and how those contributions compare to prior work is thorough. The characterization of model weight change into magnitude and consistency of updates provides an intuitive framing of the method, and their approximation of both of these these changes as a weight difference after a fixed number of model updates makes the method simple and tractable. The figures are useful when building intuition about the method, and clearly show how this method improves over prior work on very specific examples of domain shift.

**Weaknesses:**

This method strongly relies on the self-supervised objective being able to capture similarity and difference between train and test domains. They show that this is an effective measure for many traditional domain adaptation benchmarks, where there is a strong visual distinction between the target and source (major shifts in color or background for digit classification, for example). This type of domain shift, where there are clear visual differences that can be easily captured by a self-supervised objective, is unlikely to be as beneficial for performance estimation for both (1) domain shifts that are visually similar, but where the subpopulation shift of categories of interest leads to lower performance, and (2) domains where self-supervised objectives fail to build useful representations vs supervised ones, for example domains with fine-grained categories (see https://arxiv.org/abs/2105.05837)

**Questions:**

The lack of demonstration of the method on more diverse or possibly more realistic types of domain shift does not render the value of the method useless, as there are many domain shift problems that are mainly characterized by visual shift or where visual shift is a strong component of performance degradation (like the shifts seen between MNIST and USPS). I would have liked to see a more nuanced discussion of what types of signal or shift this method is well-posed to capture and for what types of domain shift it is not as well-posed. This could be demonstrated with performance estimation results on more realistic domain shift benchmarks that capture the types of complex changes between “domains” seen in the real word, like the WILDS benchmark. I think this would improve the clarity of the claims the authors are making, and help possible practitioners determine whether they should rely on this performance estimation method in practice. Similarly, some discussion or formalization of what types of shift may cause maximal weight change with their chosen self-supervised objective, and why, would also strengthen the paper.

---

> ### Author Response · Authors · 2023-11-23
>
> We appreciate the reviewer's insightful feedback.
>
> In this paper, we primarily focus on distribution shifts across datasets in real-world scenarios. While the distribution shift from the MNIST to USPS datasets may not be extensive, the transition to SVHN is significant, given that SVHN comprises real photos of street view house numbers, while MNIST consists of grayscale, hand-written digits on a white background. This substantial shift is reflected in the observed drop in classifier accuracy, making the SVHN accuracy more challenging to predict. In the COCO setup, all test sets belong to different datasets, representing a natural shift as they consist of photos collected by various parties in different situations. These datasets only share the categories. The shift exists in background colors, scenery, the object's position within the picture, and other factors.
>
> We acknowledge the reviewer's interest in more structured shifts, where data shifts are based on specific components such as temporal or spatial ones. To address this, as suggested by the reviewer, we conducted additional experiments on three datasets from the WILDS benchmark:
>
> - The **Camelyon17** dataset, consisting of patches from whole-slide images indicating potential metastatic breast cancer, exhibits shifts in data collection and processing between hospitals.
> - The **iWildCam** dataset, focusing on wildlife monitoring, showcases shifts in illumination, viewing angle, background, and vegetation across camera traps, with train-test splits based on trap locations.
> - The **fMoW** dataset, categorizing satellite images into 62 classes, demonstrates shifts based on image capture years, with training before 2012 and testing after 2016.
>
> The results for the main baselines are summarized in the table below, where the values represent the absolute error between the predicted and ground truth accuracy. MAE encodes the mean absolute error across the datasets from the WILDS benchmark. In addition to the main baselines, we included ALine [1] and Nuclear Norm [2].
>
> | | Camelyon | IWildCam | FMoW | MAE|
> | ------- |---------- |----------| --------- | ----- |
> |Entropy ($\tau$ = 0.1) |54.52 | 0.94 | 25.34 | 26.94|
> |Entropy ($\tau$ = 0.3) |19.31 | 21.41 | 0.45 |  13.72|
> | ATC | 11.31 | 1.66 | 2.37 | 5.11 |
> | Nucl | 14.45 | 7.13 | 4.05 | 8.54 |
> | ALine | 5.47 | 4.95 | 1.30 | 3.91|
> | FID | 1.69 | 8.19 | 4.87 | 4.92 |
> | COT | 2.95 | 11.68 | 0.32 | 4.98 |
> | **Eucl D(our)** | 1.44 | 3.88 | 0.6 | **1.97** |
> | GTruth | 72.91 | 67.69 | 52.90 | |
>
>
> The results indicate that our method consistently outperforms the other approaches. Please note that we used the same basic augmentations to create the meta-set, demonstrating that our approach is not sensitive to the augmentation strategies employed.
>
> We have included the results for the Wilds benchmark in the the appendix of the updated paper, along with technical details regarding the training and fine-tuning of the corresponding networks.
>
> [1] Christina Baek, Yiding Jiang, Aditi Raghunathan, Zico Kolter. "Agreement-on-the-line: Predicting the performance of neural networks under distribution shift". Advances in Neural Information Processing Systems, 2022.
>
> [2] Weijian Deng, Yumin Suh, Stephen Gould, Liang Zheng. "Confidence and Dispersity Speak: Characterizing Prediction Matrix for Unsupervised Accuracy Estimation". International Conference on Machine Learning, 2023

---

### Official Review · Reviewer_j3d2 · 2023-11-02

**Soundness:** 2 fair
**Presentation:** 2 fair
**Contribution:** 1 poor
**Rating:** 3
**Confidence:** 5

**Summary:**

This study challenges the standard in performance prediction models that utilize activation or performance metrics, highlighting their diminished accuracy when facing domain shifts. This work suggests employing a weight-based metric, observing the variance in the model's weights pre- and post-fine-tuning with a self-supervised loss—specifically, the entropy of the network's predictions. The premise is that minor weight adjustments post-fine-tuning correlate with target data similarity to the source domain, suggesting higher confidence in predictions.

**Strengths:**

+ This motivation is sound. Existing methods aim to explore a proxy that can reflect the model accuracy on unlabeled test sets. They typically use feature representations or model outputs. This work explores the weights information.

+ It seems the authors try very hard to build up the experimental setups, including dataset creation, and train various models, and build up baselines.

**Weaknesses:**

- *Lack of Novelty*: The method proposed in this manuscript closely resembles the technique described in "ProjNorm" by Yu et al. (2022). While the authors have introduced Shannon entropy as a means to differentiate from the aforementioned work, this change appears marginal and does not substantially deviate from the core idea presented in ProjNorm. The application of Shannon entropy in this context does not seem to provide a significant methodological improvement or lead to new insights, as the primary concept and application remain largely unchanged.

- *Limited Experimental Setup*: The experimental framework, as it stands, is limited in scope. For the work to be comparative and relevant, inclusion of standard datasets such as CIFAR-10 and ImageNet-1K is essential. These datasets are benchmarks in the field and are used across recent literature, providing a common ground for comparing innovative approaches. Furthermore, the paper should benchmark against recent studies, particularly those concerning prediction under distribution shift and unsupervised accuracy estimation. I recommend the authors examine works such as "Predicting out-of-distribution error with the projection norm," "Agreement-on-the-line: Predicting the performance of neural networks under distribution shift," and "Confidence and Dispersity Speak: Characterising Prediction Matrix for Unsupervised Accuracy Estimation."

**Questions:**

- Please see the above weakness and pay attention to ProjNorm which already reported the same idea. Please consider the current methods and especially the setups as so to make experiment more convincing.

---

> ### Author Response · Authors · 2023-11-23
>
> We thank the reviewer for their insightful evaluation. Below, we address their main concerns.
>
> The reviewer listed several papers, and we aim to clarify why they were not included in the primary comparison analysis.
>
>   - Agreement-on-the-line: Predicting the performance of neural networks under distribution shift [1]. This paper employs a phenomenon called "agreement-on-the-line," - a strong linear correlation between in-distribution and out-of-distribution agreement among neural network classifiers. This method requires a large number of trained networks, which are only available for the most common and well studied datasets (e.g. 467 models for CIFAR-10, 269 models for Camelyon17-Wilds, see Appendix 4 of [1]). This does not meet the requirements of the datasets we are targeting in our paper.
>
> - The papers, 'Confidence and Dispersity Speak: Characterizing Prediction Matrix for Unsupervised Accuracy Estimation' [2] and 'Predicting out-of-distribution error with the projection norm' [3], may appear similar at first glance, but they target different tasks. Specifically, they focus on the correlation analysis between their respective measures and out-of-distribution (OOD) accuracy. Despite strong correlations, as seen in our work with Rotation accuracy, it is evident that this does not directly translate into accurate performance estimation.
> While the Nuclear Norm measure proposed in [2] can be directly used for performance estimation, as it is normalized between 0 and 1 and has a direct relation to accuracy as a measure of dispersity and confidence of the network, the Projection Norm value is not directly linked to the accuracy value itself. Our supplementary material, Table 6, further demonstrates that adapting the Projection Norm to our task through meta-dataset creation leads to inaccurate performance predictions. As detailed in Section 3.2, the difference between weights reflects both the magnitude and consistency of network updates. Unsupervised entropy minimization scales the magnitude of the gradient values based on network uncertainty for each sample, while ProjNorm relies entirely on the correctness of the pseudo-labels.
>
> The reviewer suggested including the CIFAR10 and ImageNet-1K datasets in our pipeline. The results for CIFAR10 are presented in Table 3 of our main experiments. As suggested by the reviewer, we performed additional experiments on ImageNet1K – ImageNet V2 setup. We also added 3 additional datasets from the WILDS benchmark. Finally, we added the baselines proposed by the reviewer, namely Aline[1] and Nucl[2]. The results  are summarized in the tables below, where the values represent the absolute error between the predicted and ground truth accuracy. MAE depicts the mean absolute error across the datasets from the WILDS benchmark.
>
> The results on all three datasets show that our method outperforms the other approaches. We will add all these experiments in the final version of our paper.
>
> |              | Camelyon | IWildCam  | FMoW | MAE|
> | -------   |----------      |----------| ---------      |   ----- |
> |Entropy ($\tau$ = 0.1) |54.52 | 0.94 | 25.34 | 26.94|
> |Entropy ($\tau$ = 0.3) |19.31 | 21.41 | 0.45 |  13.72|
> | ATC     |       11.31  |    1.66 |        2.37      |  5.11 |
> | Aline [1] |     5.47  |     4.95 |        1.30      |   3.91|
> | Nucl [2]     |       14.45 |   7.13  |       4.05       |  8.54 |
> | FID      |        1.69  |     8.39 |	       4.89      |  4.99 |
> | COT    |         2.95   |   11.68 |     0.32  |       4.98 |
> |  EuclD(our) | 1.44   |     3.88  |     0.6    |     1.97 |
> | GTruth |      72.91    |  67.69  |   52.90 |              |
>
>
> |              | ImageNetV2 |  CIFAR10.1 |
> | -------   |----------      |   ---------------    |
> |Entropy ($\tau$ = 0.1) |13.47 | 3.25 |
> |Entropy ($\tau$ = 0.3) |14.53 |  10.90|
>  |ATC     |       0.97 |    1.50  |
> | Aline [1] |     2.06  | 1.11 |
> | Nucl [2]   |       2.40    |  5.02    |
> | FID       |      5.73       |  8.90  |
> |COT	    |   2.46  |0.10      |
> |  EuclD(our) | 0.58  |  0.98  |
> | GTruth |   71.03   |  88.65  |
>
>
>
> [1] Christina Baek, Yiding Jiang, Aditi Raghunathan, Zico Kolter. "Agreement-on-the-line: Predicting the performance of neural networks under distribution shift". Advances in Neural Information Processing Systems, 2022.
>
> [2] Weijian Deng, Yumin Suh, Stephen Gould, Liang Zheng. "Confidence and Dispersity Speak: Characterizing Prediction Matrix for Unsupervised Accuracy Estimation". International Conference on Machine Learning, 2023
>
> [3] Yaodong Yu, Zitong Yang, Alexander Wei, Yi Ma, Jacob Steinhardt. "Predicting Out-of-Distribution Error with the Projection Norm". International Conference on Machine Learning 2022

---

### Official Review · Reviewer_xDdC · 2023-11-02

**Soundness:** 3 good
**Presentation:** 3 good
**Contribution:** 2 fair
**Rating:** 5
**Confidence:** 4

**Summary:**

The paper proposes a method for estimating the generalization performance of deep neural networks under distribution shifts. The key contribution is the analysis of network weights to estimate generalization. The authors show that the distribution shift between the source and target datasets can be captured by the difference in network weights. Insipire by this, they propose a weight-based approach that estimates the performance of the network from the degree of weight changes incurred by fine-tuning the network on the target dataset with an unsupervised loss. Experimental results show that the proposed method is effective in estimating generalization performance on three image classification datasets with different backbones.

**Strengths:**

1. In addition to activation-based approaches and performance-based approaches, the paper proposes a weight-based approach for model accuracy prediction. This approach makes sense and is somewhat novel to me. The method is simple and straightforward.

2. The performance of the proposed approach is good compared to performance-based and activation-based methods.

3. The paper writing is clear and easy to understand.

**Weaknesses:**

1. The paper presents a reasonable alternative to performance-based and activation-based accuracy prediction methods. However, the conclusion is not much surprising to me, since utilizing the change of weights has already been a common approach, showing effectivenss in extensive unsupervised and semi-supervised learning literature. Besides this point, I didn't see many significant flaws in this paper. Therefore, the paper is kind of on the 'borderline' to me. My initial rating is borderline accept.

2. An experiment of $N$-model accuracy ranking would make the conclusion stronger. The proposed only evaluates one model compared against a GT accuracy. However, the MAE of accuracy is hard to interpret, e.g., would 1% on one dataset be better than 3% on another one? It's more meaningful to rank accuracies of $N$ models trained on the same dataset then compare the predicted ranking list to GT ranking list. This evaluation may be a better metric than MAE.

3. Some related work that investigated model weight changes for SSL are expected be discussed, e.g., MeanTeacher [1] and Temporal Ensembling [2].

[1] Tarvainen, Antti, and Harri Valpola. "Mean teachers are better role models: Weight-averaged consistency targets improve semi-supervised deep learning results." Advances in neural information processing systems 30 (2017).

[2] Laine, Samuli and Aila, Timo. Temporal Ensembling for Semi-Supervised Learning. ICLR, 2017.

---------------------------------
After Rebuttal:

I thank the authors for the response. I acknowledge there are differences between this work and SSL literature. However, I have not been fully convinced that this work does not conduct model ranking evaluation because "it does not leverage the consistency in model updates". This method is essentially built upon the consistency in model updates before and after finetuning on target domain.

I partly agree with Reviewer j3d2 that this work is similar to ProjNorm: The core concepts of ProjNorm and this work are similar, using weight changes before and after finetuning for performance estimation. ProjNorm uses pseudo labels for target domain, while this work uses Shannon entropy for target domain optimization, as well as includes an additional regression step. I think the differences exist but are not that significant.

In addition, I agree with Reviewers j3d2 and PWrH that there are still some issues in evaluation including the evaluation metrics and the not-very-significant performance improvement.

Therefore, my final rating is borderline reject.

**Questions:**

See Weaknesses

---

> ### Author Response · Authors · 2023-11-23
>
> We thank the reviewer for providing valuable insights in their feedback.
>
> The reviewer mentioned SSL approaches by Antti et.al and Samuli et.al., which employ a teacher-student framework and consistency regularization to encourage the model to produce consistent predictions over time. While acknowledged for their effectiveness in enhancing network generalizability and robustness against distribution shifts, these methods fail to address the aspect of predicting the network improvement. In essence, they enhance generalizability, but are not addressing the performance prediction problem.
>
> The experimental setup outlined by the reviewer, where multiple models are accessible for each dataset, deviates from our primary objective of performance prediction. Instead, it aligns more closely with the problems of unsupervised transferability estimation/model selection. In such scenarios, our approach of fine-tuning with an unsupervised loss may not be the most suitable, as it does not leverage the consistency in model updates, similarly to the techniques employed in SSL methods. It also does not make use of the agreement  between models, as demonstrated in [1, 2]. To enhance the competitiveness of our approach within this experimental setting, it is imperative to adapt our method by minimizing computational complexity, for example, through one-shot learning. This will be a focus of our future work.
>
> [1] Yiding Jiang, Vaishnavh Nagarajan, Christina Baek, and J. Zico Kolter. Assessing generalization of sgd via disagreement. ICLR 2022
>
> [2] Ching-Yao Chuang, Antonio Torralba, and Stefanie Jegelka. Estimating generalization under distribution shifts via domain-invariant representations. International Conference of Machine Learning. ICML, 2020

---

### Meta-Review · Area_Chair_D9W9 · 2024-01-02

**Metareview:**

Summary:
The paper proposes a weight-based method to estimate the generalization performance of deep neural networks under distribution shifts. By analyzing network weights, the authors show that differences in weights capture distribution shifts between source and target datasets. The proposed approach employs unsupervised loss during fine-tuning on the target dataset and proves effective across three image classification datasets with different backbones. This study challenges traditional models based on activation or performance metrics, advocating for a weight-based metric, particularly the entropy of network predictions, which correlates with target data similarity to the source domain, ensuring more reliable predictions in the presence of domain shifts.

Strengths:
The paper presents a robust and straightforward weight-based approach for predicting model accuracy, outperforming both performance-based and activation-based methods. The motivation behind this work is well-founded, highlighting a departure from existing methods that rely on feature representations or model outputs to instead explore the informative aspect of weights, which provides a valuable perspective for accurate model predictions on unlabeled test sets.

Weaknesses:
The paper exhibits several weaknesses that merit consideration. Firstly, the lack of novelty is notable, as the approach relying on changes in weights has been a common strategy in extensive unsupervised and semi-supervised learning literature. Moreover, the method's resemblance to the technique presented in "ProjNorm" raises concerns about originality. Additionally, questions arise about the evaluation metrics with the use of Mean Absolute Error (MAE) compared with ground truth accuracy, prompting concerns about the robustness of the evaluation methodology. The experimental setup is limited, lacking experiments on standard datasets and benchmarks, and failing to compare with recent studies on prediction under distribution shift and unsupervised accuracy estimation. The absence of demonstration on more diverse or realistic types of domain shift may also hinder its broader applicability.

**Justification For Why Not Higher Score:**

While the paper demonstrates some insightful values in the target problem, the concerns and questions raised by reviewers including novelty, evaluation metrics, experiment setting etc. still exist after the rebuttal. As it stands, the current version of the paper may not be ready to be accepted to be published and a more significant revision is needed. Thus, I suggest Reject.

**Justification For Why Not Lower Score:**

N/A

---

### Decision · Program_Chairs · 2024-01-16

Reject